Predictors of tuberculosis disease in smokers: a case-control study in northeastern Malaysia

Tengku Khalid Tengku Noor Farhana 1
Wan Mohammad Wan Mohd Zahiruddin 1
Ab Samat Razan 2
Nik Husain Nik Rosmawati rosmawati@usm.my 1
1 Universiti Sains Malaysia, Department of Community Medicine, School of Medical Sciences, Health Campus , Kubang Kerian , Kelantan , Malaysia
2 Bachok District Health Office , Bachok , Kelantan , Malaysia
Adnan Mohd
Electronic publication date: 2022 Sep 6
Publication date: 2022
Volume: 10
Electronic Location ID: e13984
Received 2022 Mar 15; Accepted 2022 Aug 10
Copyright: ©2022 Tengku Khalid et al.
Copyright year: 2022
Copyright holder: Tengku Khalid et al.
License: This is an open access article distributed under the terms of the Creative Commons Attribution License, which permits unrestricted use, distribution, reproduction and adaptation in any medium and for any purpose provided that it is properly attributed. For attribution, the original author(s), title, publication source (PeerJ) and either DOI or URL of the article must be cited.
License URL: https://creativecommons.org/licenses/by/4.0/

Keywords: Tuberculosis, Smoking, Predictors, Case-control

Funding: Graduate Education Development Incentive Fund (TIPPS), School of Medical Sciences, Unviersiti Sains Malaysia This work is funded by the Graduate Education Development Incentive Fund (TIPPS), School of Medical Sciences, Unviersiti Sains Malaysia. The funders had no role in study design, data collection and analysis, decision to publish, or preparation of the manuscript.

==============================
Objective

Tuberculosis (TB) is a leading infectious disease. However, many TB cases remain undetected and only present symptoms at a late stage of the infection. Therefore, targeted TB screening in high-risk populations, including smokers, is crucial. This study aimed to determine the predictors of TB disease among the smoker population in northeast Malaysia from 2019 to 2020.

Methods

A case-control study was conducted involving smokers aged 18 years and older from health clinics in Bachok Kelantan, Malaysia. Data were collected via face-to-face interviews or telephone calls from 159 participants, randomly selected from outpatient TB records. Simple and multiple logistic regression, using R software, were used to identify the determinants of TB.

Results

Most participants were male (59.1%) and had a secondary education (56.0%). Active smokers constituted 35.2% of the group, and the mean (SD) duration of exposure to smoking was 23.9 (16.47) and 18.4 (12.84) years for the case and control groups, respectively. Being an ex-smoker (adjusted odds ratio (AOR) 6.17; 95% CI [1.55–28.32]; p = 0.013), weight loss (AOR 13.45; 95% CI [4.58–44.46]; p < 0.005), night sweats (AOR 63.84; 95% CI [8.99–1392.75]; p < 0.005) and duration of symptoms (AOR 1.02; 95% CI [1.01–1.04]; p = 0.022) were shown to be significant predictors for TB disease.

Conclusion

Four predictors of TB disease in the population of smokers were recognised in this study and should be prioritised for early TB screening and diagnosis. This may help increase TB detection, initiate prompt treatment and reduce complications among the group at risk for TB.

Introduction

Tuberculosis (TB) is the leading infectious disease globally, and it significantly impacts countries’ health and financial burden. The World Health Organization (WHO) reports that in 2020, almost 10 million people were diagnosed with TB and nearly 1.5 million died from the disease (World Health Organization, 2021). Most of these came from lower- and middle-income countries. Generally, TB cases show a declining trend, but the reduction has been slower than expected. For example, in 2018, the TB rate had declined by only two per cent relative to 2017 (World Health Organization, 2018). One of the reasons for this is that many TB cases remain undetected. In 2015, the WHO estimated that, globally, approximately 3.6 million TB cases went undetected (World Health Organization, 2015).

To enhance TB detection, the WHO has advocated a global approach to implementing a systematic TB screening programme that targets high-risk groups, including smokers, diabetic patients, people living with HIV and the elderly (World Health Organization, 2013). Smoking is considered high-risk behaviour for TB disease (Toossi, 2000; Davies et al., 2006; Mahishale et al., 2015). Few known chemicals in cigarette smoke such as formaldehyde, tar and acrolein act as irritants to the respiratory system by causing inflammatory processes and injury to the epithelium and submucosal secretory glands (Aghapour et al., 2018). These pathological changes lead to mucus hypersecretions and deficiency of ciliary oscillations. Mucus accumulation and delay in mucociliary clearance make the environments favourable for Mycobacterium tuberculosis colonisation.

In Malaysia, TB screening of smoker populations is conducted when they present themselves at the Quit Smoking Clinic (Ministry of Health Malaysia, 2016). All smokers who come for appointments at that clinic will be screened for TB disease annually or, based on symptomatic screening, will be subjected to chest X-ray examination. To date, there are no specific criteria outlining TB screening protocols for smokers, although a few studies have reported the determinants of TB disease among this group. Lin et al. (2009) report that older smokers and smokers who live in crowded homes are at greater risk for TB. Smokers who are heavy alcohol drinkers are also at higher risk for active TB disease (Jee et al., 2009). According to smoking status, a higher risk of TB was reported among current smokers, ex-smokers and passive smokers compared to non-smokers (Bates et al., 2007; Jee et al., 2009; Leung et al., 2010; Lindsay et al., 2014). Furthermore, the longer the duration of smoking and the higher the number of cigarettes smoked both increase the risk of contracting TB (Leung et al., 2004; Bates et al., 2007; Jee et al., 2009; Smith et al., 2015; Zhang et al., 2017).

Other factors shown to be related to higher risk of TB were male gender (Jee et al., 2009; Lin et al., 2009), patients with a low body-mass index (Leung et al., 2004; Saunders et al., 2017), immigrants (Pareek et al., 2016; Chan, Kaushik & Dobler, 2017), those with a low level of education and in a lower-income group (Pablos-Méndez, Blustein & Knirsch, 1997; Lin et al., 2009), those with a TB contact (Morrison, Pai & Hopewell, 2008; Saunders et al., 2017), those with previous TB disease human immunodeficiency virus and other immunosuppressed diseases (Jee et al., 2009; Said et al., 2017), patients with chronic diseases, such as diabetes, (Moss et al., 2000; Sonnenberg et al., 2005; Benito et al., 2015; Lee et al., 2016; Al-Rifai et al., 2017) and those with TB symptoms (Valença et al., 2015; Hanifa et al., 2017).

TB disease predictors in smokers need to be determined in relation to local population and background. The current practice that screens all smokers without prioritising according to predictors consumes more resources. Furthermore, failure to recognise specific risk criteria among smokers will contribute to late TB detection and treatment, and finally, would hinder the aim of ending the global TB epidemic by 2030, as stated in the UN sustainable development goal number 3. Due to the lack of local studies and data on the factors or determinants of TB disease in smokers, a cross-sectional study was conducted to determine the predictors of TB disease among the smoker population in a northeastern state of Malaysia.

Materials and Methods

Study design and participants

A case-control study was conducted in Kelantan, Malaysia, between March and December 2021. The participants were selected according to inclusion and exclusion criteria. The inclusion criteria were outpatient attendees who came for TB screening between 2019 and 2020; smokers, whether active smokers, ex-smokers or passive smokers; aged 18 years and above; and of Malaysian citizenship. Potential participants who fulfilled the study criteria were randomly selected for inclusion in either the case or control groups. However, participants with incomplete contact numbers or addresses were excluded from the study. The cases were patients diagnosed with TB disease, while the controls were patients investigated for TB disease but whose TB results were negative. The study flowchart is shown in Fig. 1.

Figure 1 The study flowchart.

The sample size was calculated for each predictor for TB disease using Power and Sample Size calculation software to compare two independent proportions. The largest estimated total sample was 168, using the proportion of current smokers among patients not diagnosed with TB disease in South Africa: 0.59 (Pinto et al., 2013). The estimated proportion was 0.82, 5% type 1 error, 80% power, ratio 1 case to 2 controls, and an additional 10% for missing data. We used a simple random sampling method to obtain 56 cases and 112 controls from the sampling frame to compare tuberculosis with non-tuberculosis participants. The participants were interviewed via telephone or face-to-face, and information was gathered on their social history, medical history, smoking history, contact or TB history, and symptom history, using a proforma guideline constructed by the researchers. Data were also retrieved from the patients’ clinical records in the health clinics, which included the patient’s medical, contact, TB and symptom histories to ensure the accuracy of the data taken directly from participants.

Operational definitions

In this study, ‘active smokers’ referred to patients who were smoking any number of cigarettes, who had smoked within the last six months or who had quit smoking less than six months before. Patients who had quit smoking for more than six months were considered ‘ex-smokers’. A ‘passive smoker’ was a patient who had never smoked but was currently living with an active smoker. The healthcare workers included the laboratory technicians, radiology technicians, medical attendants and ambulance drivers who were directly involved with patient management in any health facility. An area with high TB incidence was defined as an area or village with two or more TB cases in 2018 and 2019. Meanwhile, ‘immunosuppressed’ referred to patients with any illnesses that suppressed their immune systems, such as chronic renal failure, ischaemic heart disease, liver disease, gastrectomy patients, and cancer, or who were prescribed medications for these. Participants were considered to have been immunised with the Bacilli Calmette-Guerin (BCG) vaccine if they had a scar on their left or right upper arm. A TB contact was defined as having been exposed to an index TB case, who could be a family member, a colleague, or a friend.

Statistical analysis

All data were recorded in Microsoft Excel and analysed using R software version 3.1.6. Descriptive data were presented using mean (standard deviation (SD)) for numerical data and number of patients (percentage (%)) for categorical data. Simple logistic regression was applied to obtain each variable’s crude odds ratio (OR). The associations between TB disease and the independent variables were analysed using simple and later, multiple logistic regression analyses. A simple logistic regression analysis was applied to all independent variables to determine their association with TB disease. Variables with a p-value of less than 0.25 were included in the multiple logistic regression analysis. The forward likelihood ratio (LR), the backward LR and Enter methods were applied to obtain the preliminary final model. Multicollinearity, two-way interactions of variables, homogeneity of variance using the Hosmer-Lemeshow goodness of fit test and area under the curve were checked to ensure the model fit before presenting the final model of the study.

Ethical approval

This study was approved by the Medical Research and Ethics Committee (MREC), Ministry of Health Malaysia (NMRR-19-3325-51853) and Human Research Ethics Committee (JEPeM), Universiti Sains Malaysia (USM/JEPeM/19110813).

Results

A total of 159 participants fulfilled the study criteria and consented to participate. Table 1 presents the basic characteristics of the study participants. Overall, more than 70% (n = 94) were male, 73% (n = 116) had studied to secondary school level at least, and 87.4% (n = 139) lived in areas with low TB incidence. Approximately 98% (n = 157) of the participants had a monthly income below RM 4850. Regarding smoking status: 35.2% (n = 56) were active smokers, 21.4% were ex-smokers (n = 34), and 43.4% (n = 68) were passive smokers. The mean (SD) duration of exposure to smoke for the cases was 23.9 (16.47) years and 18.4 (12.84) years for the control group. Only 6.3% (n = 10) of participants had been previously diagnosed with TB. The majority of participants had a history of cough (86.2%, n = 137), and the median (IQR) duration of symptoms was 30 (76) days for the cases and 7 (11) days for the control group.

Table 1 Characteristics of participants in Bachok, Kelantan (n = 159).

Variable	Case (n = 45)	Controls (n = 114)	
	n	(%)	Mean	(SD)	n	(%)	Mean	(SD)	
SOCIODEMOGRAPHY									
Age, year			48.5	(17.77)			44.2	(15.33)	
Male	32	(71.1)			62	(54.4)			
Marital status									
Never married	10	(22.2)			23	(20.2)			
Married	33	(73.3)			82	(71.9)			
Divorced/widowed	2	(4.5)			9	(7.9)			
Crowded home, Yes	11	(24.4)			43	(37.7)			
Resided in high TB incidence area	10	(22.2)			10	(8.8)			
Work in a non-medical field	45	(100.00)			109	(95.6)			
Education status									
Primary school level	11	(24.4)			16	(14.1)			
Secondary school level	29	(64.4)			60	(52.6)			
Tertiary level/College/
University	5	(11.2)			38	(33.3)			
Monthly income									
Less than RM 4,850	44	(97.8)			113	(99.1)			
RM 4,850 –RM 10,959	1	(2.2)			1	(0.9)			
More than RM 10,959	0	(0.0)			0	(0.0)			
MEDICAL HISTORY									
Diabetic patient	14	(31.1)			30	(26.3)			
HIV patient	1	(2.2)			1	(0.9)			
Immunosuppressed, yes	9	(20.0)			13	(11.4)			
SMOKING HISTORY									
Smoking Status									
Active smoker	15	(33.3)			41	(36.0)			
Ex-smoker	17	(37.8)			17	(14.9)			
Passive smoker	13	(28.9)			56	(49.1)			
Duration of exposure to smoke, year			23.9	(16.47)			18.4	(12.84)	
TB HISTORY							
BCG vaccination, yes	45	(100.0)			103	(90.3)			
Exposed to TB index case, yes	18	(40.0)			35	(30.7)			
Past history of TB disease, yes	6	(13.3)			4	(3.5)			
TB SYMPTOM									
Cough, yes	36	(80.0)			101	(88.6)			
Any sputum, yes	24	(53.3)			78	(68.4)			
Weight loss, yes	34	(75.6)			17	(14.9)			
Night sweat, yes	22	(48.9)			1	(0.9)			
Chest pain, yes	10	(22.2)			3	(2.6)			
Loss of appetite, yes	29	(64.4)			29	(25.4)			
Fever, yes	27	(60.0)			57	(50.0)			
Hemoptysis, yes	9	(20.0)			12	(10.5)			
Duration of symptoms, daya	30	(76)			7	(11)			
Notes.

a Reported in median (IQR).

Simple and multiple logistic regression analyses were used to determine the significant determinants for TB disease. Based on simple logistic regression analysis, 10 variables were significantly associated with TB disease. Another seven variables had p-values of less than 0.25. All 17 variables were included in the multiple logistic regression analysis. Being an ex-smoker, weight loss, complaints of night sweats, and duration of symptoms were significant predictors of TB disease in the smoker population (Table 2).

Table 2 Simple and multiple logistic regression models on the factors associated with TB disease among smokers in Bachok, Kelantan (n  =  159).

Variable	Simple logistic regression	Multiple logistic regression	
	Crude OR
(95% CI)	p -value a	Adjusted OR
(95% CI)	p -value b	
Social History					
Age (year)	1.02 (1.00, 1.04)	0.128			
Gender: Male (ref: Female)	2.06 (1.00, 4.54)	0.056			
Type of residential					
Crowded home: Yes (ref: No)	0.53 (0.24, 1.14)	0.114			
High incidence area: Yes (ref: Low)	2.97 (1.13, 7.83)	0.026			
Education status					
Primary school level	5.23 (1.63, 18.94)	0.007			
Secondary school level	3.67 (1.41, 11.53)	0.014			
Tertiary level/College/ University	Reference				
Medical History					
Immunosuppressed disease: No (ref: Yes)	0.51 (0.20, 1.34)	0.162			
Past TB disease: Yes (ref: No)
Smoking History	4.23 (1.15, 17.30)	0.032			
Smoking status					
Active smoker	1.58 (0.68, 3.71)	0.291	2.94 (0.77, 12.90)	0.126	
Ex-smoker	4.31 (1.77, 10.87)	0.002	6.17 (1.55, 28.32)	0.013	
Passive smoker	Reference		1.00		
Duration of exposure to smoke (year)	1.03 (1.01, 1.05)	0.030			
Presence of TB Symptom					
Cough: Yes (ref: No)	1.94 (0.74, 4.90)	0.162			
Sputum: No (ref: Yes)	1.90 (0.93, 3.85)	0.076			
Weight loss: Yes (ref: No)	17.64 (7.76, 43.12)	<0.001	13.45 (4.58, 44.46)	<0.001	
Night sweat: Yes (ref: No)	108.09 (21.08, 1984.47)	<0.001	63.84 (8.99, 1392.75)	<0.001	
Chest pain: Yes (ref: No)	10.57 (3.04, 49.14)	0.001			
Loss of appetite: Yes (ref: No)	5.31 (2.57, 11.38)	<0.001			
Hemoptysis: Yes (ref: No)	2.13 (0.81, 5.45)	0.118			
Duration of symptoms (days)	1.03 (1.01, 1.04)	<0.001	1.02 (1.01, 1.04)	0.022	
Notes.

a Simple logistic regression.

b Multiple logistic regression: No significant two-way interactions and no multicollinearity between the independent variables. Hosmer-Lemeshow test showed p = 0.888. The Area Under the Curve is 0.94.

Discussion

This study shows a higher proportion of smokers among males than females in Bachok, Kelantan, between 2019 and 2020. This is consistent with other local and global reports indicating higher prevalence of smoking in males than in females (Jee et al., 2009; Ministry of Health Malaysia, 2015). Males tend to smoke to show that they are mature enough to be accepted in society and are able to be independent. Some males think that smoking demonstrates their masculinity to the opposite gender. Moreover, the peer pressure of wanting to be included in a group with their friends causes men to smoke more than females (Allen et al., 2016; Kodriati, Pursell & Hayati, 2018).

This study shows that four predictors were significantly associated with TB disease: being an ex-smoker (adjusted OR (AOR) 6.17; 95% CI [1.55–28.32]; p = 0.013), having weight loss (AOR 13.45; 95% CI [4.58–44.46]; p < 0.005), having night sweats (AOR 63.84; 95% CI [8.99–1392.75]; p < 0.005) and duration of symptoms (AOR 1.02; 95% CI [1.01–1.04]; p = 0.022). The odds of developing TB disease increased by 6.2% for ex-smokers compared to passive smokers. This finding is congruent with other studies reporting similar observations (Jee et al., 2009; Pinto et al., 2013; Smith et al., 2015; Zhang et al., 2017). However, a prospective study using secondary data from 2,504 newly diagnosed pulmonary TB patients from 2012 to 2013 in India, observed that in both current smokers and ex-smokers, sputum smears and cultures were likely to remain positive after two months of treatment (Mahishale et al., 2015). They hypothesised that smoking was associated with more extensive lung disease, lung cavitation and positive sputum smear and culture results at baseline. A 14-year prospective cohort study (1992–2006) in South Korea reported a contradictory finding, in which current smokers had a greater risk of TB incidence than former smokers (HR = 1.4, 95% CI [1.3–1.5]) (Jee et al., 2009).

Another review paper on epidemiological evidence from the UK, China, India, and the USA summarises the association between smoking and tuberculosis (Davies et al., 2006). They report an increase in TB case rates of between two- and four-fold in smokers who smoke more than 20 cigarettes a day. The possible proposed mechanism is the effect of nicotine, which turns off the production of TNF- α by the macrophages in the lungs, rendering the patient more susceptible to the development of progressive disease from latent Mycobacterium tuberculosis infection. However, this study did not control for other factors, particularly alcohol consumption. Smoking is also reported to have a widespread effect on lung structure, such as lung obstruction, function, clearance mechanism, and host defences, both in the lung and systemically; therefore, it could plausibly increase the incidence or worsen the prognosis for TB (Jee et al., 2009).

The current study showed that active smokers had no significant association with TB disease. A possible explanation could be that ex-smokers have poorer health-seeking behaviours than active smokers. They might feel good because they are not smoking anymore, and therefore, avoid seeking early TB screening if they are having mild symptoms. Smokers may also stop smoking when TB symptoms arise, or when a TB diagnosis is confirmed (Jee et al., 2009).

We also found a significant association between weight loss, night sweats and TB disease in smokers. Smokers with weight loss had a 13.45 times higher odds of developing TB than those without weight loss. In addition, smokers with a history of night sweats had a 63.84 higher risk of contracting TB than those without the presence of night sweats. The presence of weight loss and night sweats are considered constitutional symptoms, especially in patients with chronic diseases (Toossi, 2000), including TB. Mycobacterium tuberculosis causes an inert response and enhances tissue degradation, the secretion of protein that inhibits appetite, and promotes the proliferation of cytokines, causing the symptoms of night sweats (Toossi, 2000; Cegielski & McMurray, 2004; Chang et al., 2013). Hence, patients with chronic TB infection may present with these two systemic symptoms. However, the wide confidence interval for these variables was mainly due to the small sample size, thus limiting the inferences or estimates that could be made from these findings.

The current findings are in line with a study conducted in Cape Town, which reported a higher value of adjusted ORs for the presence of weight loss (adjusted OR = 7.2, 95% CI [2.5–20]) and night sweats (adjusted OR = 1.75, 95% CI [0.62−7.9]) (Den boon et al., 2008), indicating a high likelihood of chronic diseases, such as TB, when patients are complaining of these symptoms. This study suggests that every patient presenting with these two symptoms should be tested for the presence of TB disease. The presence of weight loss and night sweats in children is also important. This is because physicians frequently face problems in diagnosing children with active TB disease due to their inability to adequately explain their symptoms and their difficulty in cooperating with diagnostic testing, such as producing sputum for the investigation. Previous researchers have developed a list of variables, represented by specific scoring, to assess the risk of TB in children. Of these, weight loss and night sweats carry the highest scores among the criteria (Graham, 2011).

The results of this study showed that a prolonged duration of TB symptoms increases the risk of having TB disease. With every one-day increase in the duration of symptoms, the odds of having TB disease were increased by 1.02 times when other factors were adjusted for. The longer the presence of TB symptoms, the more likely a patients is to have a positive TB culture. Physiologically, prolonged TB symptoms may indicate the prolonged proliferation of Mycobacterium tuberculosis in the body, particularly in the lung tissue. This reduces the lung’s ciliary function in excreting and destroying the mycobacterium, finally increasing the risk of TB disease (Toossi, 2000; Centers for Disease Control and Prevention, 2016).

Furthermore, the longer the delay in TB diagnosis and treatment, the higher the risk of more severe lung tissue damage and of the transmission of the mycobacterium to other close contacts. A cross-sectional study was conducted among newly diagnosed pulmonary TB patients in India to assess pulmonary TB’s patients’ health-seeking behaviours (Jebamalar, Senthilkumar & Ramola, 2018). The researchers reported that the reasons for delays in seeking care were the hope of self-resolution, financial constraints, fear of the diagnosis, incompatible timings and poor health-care accessibility. The time duration of having TB symptoms has been considered a crucial variable in calculating TB risk scores (Graham, 2011; Pinto et al., 2013; Hanifa et al., 2017; Jebamalar, Senthilkumar & Ramola, 2018). A significant score was given to those with a longer duration of TB symptoms, particularly when they had two or more weeks history of TB symptoms when they presented themselves.

A small sample size was one of the limitations of this study, because it limits the findings regarding the association of other variables with TB disease and results in a wide 95% CI for the adjusted odds ratio. In addition, this study was conducted in the years 2020 and 2021, during which the COVID-19 pandemic significantly affected the data collection process. Within this period, people’s movements were limited, and the number of patients who attended health clinics in Bachok, Kelantan was remarkably reduced, with many patients choosing not to attend the health clinic if they had minor illnesses, due to the fear of being infected with COVID-19.

Conclusion

This study demonstrated that significant predictors for TB disease among smokers were being categorised as an ex-smoker, losing weight, having night sweats and a longer duration of TB symptoms. Delineating these TB predictors in smokers will guide health authorities to design better and more comprehensive plans for a national TB control programme by formulating criteria for TB screening in the smoker population. A specific protocol for TB screening of smokers could be implemented in Quit Smoking Clinics and could be expanded to all outpatient departments (OPD) of primary care facilities. Consequently, early TB diagnosis and treatment in groups at risk and TB transmission prevention in the wider community would be achieved.

We would like to express our gratitude to the Ministry of Health Malaysia for their permission to conduct and publish this study.

Additional Information and Declarations

Competing Interests

Author Contributions

Human Ethics

Ethics

Field Study Permissions

Data Availability

The authors declare there are no competing interests.

Tengku Noor Farhana Tengku Khalid conceived and designed the experiments, performed the experiments, analyzed the data, prepared figures and/or tables, authored or reviewed drafts of the article, and approved the final draft.

Wan Mohd Zahiruddin Wan Mohammad conceived and designed the experiments, authored or reviewed drafts of the article, and approved the final draft.

Razan Ab Samat performed the experiments, authored or reviewed drafts of the article, and approved the final draft.

Nik Rosmawati Nik Husain conceived and designed the experiments, authored or reviewed drafts of the article, and approved the final draft.

The following information was supplied relating to ethical approvals (i.e., approving body and any reference numbers):

This study has gained ethical approval from both the Human Research and Ethics Committee (HREC) of Universiti Sains Malaysia and Malaysia Research and Ethics Committee (MREC) Ministry of Health Malaysia.

The following information was supplied relating to ethical approvals (i.e., approving body and any reference numbers):

Human Research Ethics Committee (HREC), Universiti Sains Malaysia approved this work.

The following information was supplied relating to field study approvals (i.e., approving body and any reference numbers):

Malaysia Research and Ethics Committee (MREC) Ministry of Health Malaysia approved this work to be done in Kelantan.

The following information was supplied regarding data availability:

The data is available at Zenodo: Nik Rosmawati Nik Husain. (2022). Raw data-Predictors of tuberculosis in smokers [Data set]. Zenodo. https://doi.org/10.5281/zenodo.6349779.

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
