# Peer review of "Predictors of tuberculosis disease in smokers: a case-control study in northeastern Malaysia"

_PeerJ, doi:10.7717/peerj.13984_

## Round 0.1 · original submission · Major Revisions

Reviewers have raised some serious concerns and shortcomings in the study. MAJOR revision is suggested, which requires substantial and thorough revision to appreciate the quality of the manuscript. Moreover, thorough English editing is required. Please revise the manuscript taking help from a colleague who is proficient in English and familiar with the subject matter, who can review your manuscript, or contact a professional editing service to review your manuscript. Revise and resubmit accordingly.

·

Basic reporting

Thank you, dear authors, for coming up with an excellent public health concern. However, the following issues have been addressed before being considered for publication.
Overall, the paper needs language editions.
The background is sufficient but it needs language and content editions.
Table 2 needs revision, It is good to include the AOR.

Experimental design

No comment

Validity of the findings

The validity of the finding is not questionable but it needs a revision or strong justifications for including the TB symptoms since it is observed in the result as well through a very wide confidence interval.

Additional comments

Comments
Thank you, dear authors, for coming up with an excellent public health concern. However, the following issues have been addressed before being considered for publication.

Abstracts
What do you mean by Ex-smokers? On page one line 29
Introduction
Generally, the introduction needs some improvements; there ate some concepts that need citations e.g line 43 …
Line 41 you stated WHO reported in 2022…… is it the 2022 TB report, I think it is not released. The figures taken from this paragraph are I think from the 2018 report.
Methods
How do you select the study participants? The sampling procedures are not clearly documented.
Where did you get the data collection tools? Who collects the data? How do you maintain the quality of the data? Indicate clear.
Line 135, what do you mean by univariable analysis? I think you are talking about bivariable analysis.
Result
Table one says clinical characteristics, but there are sociodemographic characteristics. Better to split it into two tables or make the title descriptive.
Line 165-167 the two sentences were written two times.
Table 2 why did you omit recording the adjusted odds ratio in some variables? Even if they are not significant it is good to write the AOR?

Variables like night sweat and weight loss are the symptoms, which are used for Tb screening, and likely all participants might have these symptoms. That is why in table night sweat has a wide confidence interval between 8.99 & 1392, we cannot say this estimate is an estimate. This is also observed in weight loss. It is good to fit the model by removing the TB symptom as a variable.
Discussion
The first paragraph in your discussion is not your objective? Your objective is predictors of TB among smokers?

Reviewer 2 ·

Basic reporting

No comment

Experimental design

No comment

Validity of the findings

No comment

Additional comments

This paper provides additional beneficial information to strengthen the TB control program in Malaysia - focusing on active TB detection among the high-risk groups in the primary care settings. However, these points need clarification:

- What was the category of the study population? (Urban, suburban, rural?) Was there any urban poor? Studies worldwide suggest that socioeconomic status is a strong determinant of health-seeking behavior.

- As stated in the literature, migrants are at high risk of contracting TB. Was the migrants/non-citizen excluded from this study?

- What is the Quit Smoking Clinic's workload/burden/enrolment rate (QSC) in Kelantan/Northeastern or Malaysia? If the enrolment is low, we might as well screen all QSC clients. Thus, the findings (predictors) should apply to the Outpatient Department (OPD) of primary care facilities in general since most of them might turn up in OPD with different ailments or are already part of the NCDs registry.

---

## Round 0.2 · accepted · Accept

Manuscript is significantly improved by the authors and now can be accepted in its current form.

·

Basic reporting

No comment

Experimental design

No comment

Validity of the findings

No comment

Additional comments

The authors addressed my concerns adequately.

Reviewer 2 ·

Basic reporting

Good and appropriate

Experimental design

Appropriate. Carried out according to the aim and scope of the journal.

Validity of the findings

Appropriate

Additional comments

The author clarified the issues raised. The findings have scientific merit that can be applied to improve TB Case detection rate and increase early detection.